# Cenozoic pelagic accumulation rates and biased sampling of the deep-sea record

Johan Renaudie[1] and David B. Lazarus[1]

[1]Museum für Naturkunde, Leibniz-Institut für Evolutions- und Biodiversitätsforschung, Invalidenstrasse 43, 10115 Berlin, Germany

**Correspondence:** Johan Renaudie (lecoryphee@googlemail.com)

**Abstract.** Global weathering is a primary control of the Earth's climate over geologic time scales: converting atmospheric $p_{CO_2}$ into dissolved bicarbonate; with carbon sequestration by marine plankton as carbonate and organic carbon on the ocean floor. The accumulation rate of pelagic marine biogenic sediments is thus an indication of weathering history. Previous studies of Cenozoic pelagic sedimentation have yielded contrasting results, though most show a dramatic rise (up to 6 times) in rates over the Cenozoic. This contrasts with model expectations for approximate steady state in weathering, $p_{CO_2}$, and sequestration over time. Here we show that the Cenozoic record of sedimentation recovered by deep-sea drilling has a strong, systematic bias towards lower rates of sedimentation with increasing age. When this bias is removed accumulation rates are shown to actually decline by ca 2 times over the Cenozoic. When accumulation area however is adjusted for changes in available deposition area, global sediment flux to the deep-sea is shown to have nearly doubled at the Eocene-Oligocene boundary, but was otherwise essentially constant. Compilations of other metrics correlated to sedimentation rate (e.g. productivity, biotic composition) also must have a strong age bias, which will need to be considered in future paleoceanographic studies.

## 1 Introduction

### 1.1 Cenozoic weathering

The rate of weathering of the Earth's surface over geologic time is a fundamental characteristic that determines the rate at which many crucial geologic cyles function, and affects equilibrium levels of many key chemical components, including levels of atmospheric $CO_2$ (Berner et al., 1983; Caves Rugenstein et al., 2019). A very large and complex literature has accumulated on the geologic history of weathering: while we will not present here a full review of that literature, we will summarize that which is most directly related to our own study of rates of accumulation of Cenozoic pelagic sediment, and its use as a proxy for rates of Cenozoic weathering. In addition to more or less pure modeling studies which examine how different proposed factors might have acted over geologic time (e. g. Southam and Hay, 1977), studies attempting to use proxies to reconstruct the actual weathering history over the Cenozoic can very broadly be divided into two general types - those using geochemical proxies, often isotope ratios in marine sediment thought to be sensitive to rates of chemical weathering; and those using measurements of global sedimentary accumulation derived from weathering's products.

Most of these studies also work within a framework of two competing end member hypotheses – one, that rates of weathering have been nearly constant over time; or two, that rates of weathering have changed significantly over the Cenozoic, and usually increasing towards the recent. In sediment accumulation studies both views start with the general observation that on a global scale preserved amounts of sedimentary material (both chemical weathering derived, and clastic) decline with increasing geologic age (e. g. Garrels and MacKenzie, 1971). The steady state hypothesis interprets this pattern of decline as an artifact of increasingly poor preservation of older sediment (Garrels and MacKenzie, 1971), while the change hypothesis interprets the pattern as being primarily driven by changing causal factors of weathering and erosion, such as the pronounced cooling and increase in tectonic activity over the course of the Cenozoic. More general reviews can be found in Garrels and MacKenzie (1971); Kump et al. (2000); Ruddiman (2013); Penman et al. (2020); Hilton and West (2020).

## 1.2 Geochemical approaches

Cenozoic weathering reconstructions from geochemical proxies include those based on isotopes of Sr, Os, Be, and Li, among others. A variety of conclusions have been drawn, ranging from near steady state, or at most modest increases in weathering over the Cenozoic (e. g. Kump and Arthur, 1997; Willenbring and von Blanckenburg, 2010; Vigier and Goddéris, 2015; Willenbring and Jerolmack, 2016; Caves Rugenstein et al., 2019; Katchinoff et al., 2021) to substantial increases, or at least strongly varying rates of Cenozoic weathering (e. g. Raymo et al., 1988; Delaney and Boyle, 1988; Peucker-Ehrenbrink et al., 1995; McCauley and DePaolo, 1997; Lear et al., 2003; Misra and Froelich, 2012; van der Ploeg et al., 2019). There are however many general limitations that studies of this type must deal with. The numerous controls in nature that generate the isotope ratios, or other geochemical proxies that are measured are only partially known, and past behavior many of the processes that are known to be of importance are still not well constrained (see e. g. von Strandmann et al., 2020, review for Li). In some cases poorly constrained parameters in studies are evaluated by considering a range of possible values, but occasionally unknown parameters are simply set to rather arbitrary values, or even assumed, despite the lack of any supporting evidence, to be constant over geologic time. Many study parameter calibrations are also based on geographically limited data from modern measurements; nor is it clear if these short time baseline values (often being based on a few decades of observational data, or from the last few thousand years geologic record, e.g. from Holocene sediments) can be directly scaled to the much longer million year plus time periods of the Cenozoic. Further, most studies focus on only one or a few aspects of the very complex earth system - the potential effects of changing uplift; weatherability; deep ocean temperatures; or reverse weathering, to mention just a few. As these individual works convincingly show that these different aspects are all potentially important in controlling weathering history, a realistic reconstruction of weathering history should in principle include the all of these processes, as well as realistic histories of the large number of controlling parameters (Delaney and Boyle, 1988); but to our knowledge such comprehensive studies are only beginning to be constructed (e. g. Caves Rugenstein et al., 2019). As important as these studies are in providing a range of valuable constraints on weathering history, and in evaluating the many processes that affect weathering and proxies for it, additional studies that more directly measure the output of weathering, e.g. via authigenic or biogenic marine sediment formation, are also essential to understand the evolution of Cenozoic weathering.

## 1.3 Prior studies of sediment accumulation

The products of weathering are dissolved ions, transported via rivers and groundwater from land to sea; or in deep ocean marine water percolating through the ocean crust. These ions are then removed from marine waters via the formation of (mostly clay) minerals by direct authigenic processes (called reverse weathering), or by biologic capture of the ions to form shells and skeletons by organisms either in shallow benthic, or open ocean pelagic environments. The relative proportions of ion removal by minerals, shallow biota, or pelagic plankton remain controversial. While originally most removal was thought to be by plankton, more recent work has noted significant removal by shallow marine biotas, and reverse weathering processes. The proportion of the weathering ions thought to be removed by shallow water processes differs between calcium, at ca 50% (Milliman, 1993) and silica, at ca 30% (Tréguer et al., 2021). Rates of accumulation of Cenozoic sediment in shallow marine areas are difficult to measure, and generally cannot distinguish between abundant clastic sediment, a product of erosion, and authigenic-biogenic sediment derived from weathering ions. Such studies (Olson et al., 2016) have however also generally found increased rates of sediment accumulation over the Cenozoic, although increases occurred mostly only in the last few Myr. This pattern of increase, but mostly in the late Neogene, is also seen in clastic sedimentation in Cenozoic continental sedimentary basins (Molnar, 2004).

## 1.4 Cenozoic pelagic sedimentation

Pelagic sediments thus contain at least half of all weathering export, and are usually relatively free of clastic erosional sedimentation products. The rates at which pelagic sediments accumulated over the Cenozoic are therefore probably the best single direct measure of Cenozoic weathering output. As however rates of pelagic sediment accumulation vary enormously by location (Lisitzin, 1972), most studies are based on global compilations which are assumed to largely compensate for geographic variation. Early studies were based on a compilation by Davies et al. (1977) of sediment volumes by geologic age (approximately sub-epoch resolution) from the very first phase of deep-sea ocean drilling (335 Sites from DSDP Legs 1-32). An updated dataset was created by Sloan (1985) (summarized in Hay et al., 1988) that covered up to Legs 79. This improved dataset, with not only contains many more, and more continuously cored sites, but also have an age resolution limited to 5 Myr intervals, was used in several subsequent studies. Analyses of these datasets by the original authors (e. g. Davies et al., 1977; Whitman and Davies, 1979; Hay et al., 1988) were primarily used to investigate total inventories of deep-sea sediment and mass accumulation over time rather than as proxies for weathering, e.g. via computing rates of accumulation per unit area. The strong exponential decay curve shape to the total sediment mass remaining per time interval was attributed to factors that destroy the geologic record over time, and in particular for pelagic sediments erosion and subduction of ocean crust. Subduction loss of sediment volume was explicitly calculated via sea floor age data (e. g. Hay et al., 1988). Erosion was examined via the age distribution of hiatuses, as indicated by the absence of biostratigraphic zones, and found to increase with increasing age, similar to patterns reported on land (Moore Jr and Heath, 1977). This in turn was used to adjust the Cenozoic carbonate accumulation curve of Davies et al. (1977) to one showing only a modest net increase, although with very large oscillations on ca 10 Myr scales (Davies and Worsley, 1981). It should be noted that subsequent citations of these early works frequently refer

to the total mass per unit time curves of Davies et al. (1977) or Hay et al. (1988), which have not been corrected for subduction loss of record.

Possible systematic deline in Cenozoic sediment accumulation rates per unit area with increasing age, as an alternate explanation for the observed decay curve were only sketchily addressed, and instead it was often implicitly assumed that initial accumulation rates per unit area have been appoximately constant, following the model of Garrels and MacKenzie (1971). Some studies did note changes in accumulation rate, but primarily those deviating from the general fitted exponential decay curve. The Eocene and mid Miocene-Recent intervals for example were noted to have accumulation rates several times higher than the Paleocene and Oligocene intervals in-between (Davies et al., 1977; Davies and Worsley, 1981). Accumulation rates per unit area per time interval were only reported, to our knowledge, from the works of Davies and Worsley (1981), and of Sloan (1985) in Hay et al. (1988) in tabular form, though in the latter study they were not discussed further.

Research on marine sediment accumulation rates became less active subsequently as the use of geochemical proxies to study mass fluxes in the earth system increased. Proponents of near steady state rates of chemical weathering have again suggested that the pattern of increase over time seen in these earlier deep-sea sediment compilations is indeed a bias due to erosion/hiatuses, noting the widespread existence of declining rates of sedimentation with increasing geologic age in multiple studies, and citing Sadler (1981) model as the explanation (e. g. Willenbring and von Blanckenburg, 2010; Willenbring and Jerolmack, 2016).

As noted above, some of these studies' comparisons of rates of weathering, derived from geochemical proxies, to rates of sediment accumulation have used, or included, rates of terrigenous (clastic) sedimentation, although clastic sediments, being derived by mechanical erosion, are not direct products of chemical weathering. While there is some evidence that rates of erosion and chemical weathering are correlated (Riebe et al., 2004; West et al., 2005), changes in the ratio may well have occurred over the Cenozoic (Misra and Froelich, 2012; Vigier and Goddéris, 2015), calling into question the validity of such comparisons. Further, rates of sediment mass by age (mass per unit time) as cited from early studies are, as noted in these source studies, not directly comparable to rates of sediment accumulation (mass per unit area per unit time) due to subduction of older crust. Rates of sediment accumulation per unit area in the compilation of Sloan (1985)(in Hay et al., 1988), show a much more modest increase over time (Figure 1) although there are many limitations in these nearly 40 year old datasets, including poor drilling recovery of individual sections, limited biostratigraphic data, and others. New analyses of pelagic sedimentation are thus important. Westacott et al. (2021) provides an example. Using substantially improved deep-sea sediment databases (Cramer et al., 2009; Lisiecki and Raymo, 2005), they observed that per unit area accumulation rates have seemed to increase dramatically over Cenozoic time - in their study by a factor of nearly 6X. This seems to justify the view that sediment accumulation rate data give a fundamentally different result than many geochemical proxy studies; and thus also the importance in reconciling the differences, such as by the suggestion of a major erosional/temporal scaling meaurement overprint (i. e. a Sadler effect; Sadler, 1981; Willenbring and von Blanckenburg, 2010; Willenbring and Jerolmack, 2016).

The 'Sadler effect' (Sadler, 1981) is based on a shallow water sedimentation model of extensive hiatuses, a hierarchy of 'ever more and longer hiatuses' as measurement time interval increases, and decreasing age measurement resolution with increasing geologic age (Sadler, 1981). These factors work together to yield apparent decreasing SAR in many types of stratigraphic

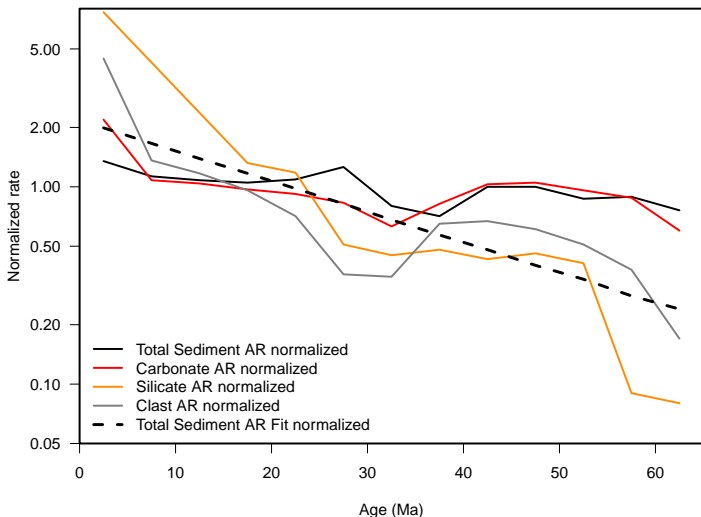

**Figure 1.** Relative trends in sediment accumulation from data given in Hay et al. (1988), normalized to each series' mean value to show similarities and differences in magnitude and patterns of change. Note log scale on vertical axis. The accumulation rates (in our study, corrected for the effect of sediment loss via subduction) for different sediment types (carbonate, biogenic silica and clastic) are shown, as well as the original paper's fitted exponential function - uncorrected for subduction - of total sediment mass vs geologic age (dashed line). This latter is the one most frequently cited in other studies. AR: accumulation rate.

sections with increasing age and time scale of measurement (Sadler, 1981). In the deep-sea however, with usually continuous (on scales > 1 yr) if variable rate rain from surface productivity and a much lower energy benthic sediment erosion regime, one would not expect to see strong Sadler effect. Sections in paleoceanographic studies are frequently reported as nearly complete at biostratigraphic zone to orbital tuning resolution (ca 500-20 kyr), typically with only a few, limited duration hiatuses. Pelagic microfossil biostratigraphic zones are also similar in resolution over the entire Cenozoic (Vandenberghe et al., 2012; Hilgen

et al., 2012). Nor was there an observed effect in the original study of Sadler in his (limited) deep-sea data on any time scale greater than ca 0.5 Myr (Sadler, 1981). Lastly, no detectable Sadler effect was seen in a recent analysis of global Neogene pelagic organic carbon accumulaton rates (Li et al., 2023). It is thus important to test if the Sadler effect is actually responsible for the observed several fold increase in Cenozoic pelagic sedimentation rates (e. g. Westacott et al., 2021). If the deep-sea pelagic sedimentary record is not (strongly) biased by hiatus-driven measurement scale artifacts, then the major increase in

apparent rate over Cenozoic means either that geochemical proxies studies reporting constant rates of weathering are somehow wrong and rate has changed exponentially, or that some other factors are biasing the Cenozoic deep-sea record.

Here we examine 1) hiatus biases affecting apparent Sediment Accumulation Rates (hereafter SAR); 2) whether SAR (excluding hiatuses) increased through time, 3) demonstrate a new mechanism - drilling bias - that is responsible for the apparent decline in SAR, 4) apply this insight to determine Cenozoic sediment fluxes, and 5) briefly discuss the broader implications of

140 drilling bias for paleoceanographic research.

## 2  Workflow

This study consists of several subanalyses carried out sequentially, each building on the results of the prior sub-analysis. We summarize here this sequence of analysis before going into the details of their methods and results. This study's workflow is summarized in Figure 2.

### 2.1  Impact fo hiatus distribution on SAR

We first show that hiatuses in deep-sea pelagic sediment sections are relatively infrequent and mostly of short duration, nor do they increase in magnitude with increasing geologic age (see Figure 3). Thus the apparent decline of accumulation with age could not be due to biassing by the cumulative effect of common, and with increasing scale of measurement (see Figure 4), ever larger hiatuses on the calculation of apparent accumulation rates (as had been speculated by some prior authors). This result suggests that another cause, or set of causes must be responsible.

### 2.2  By-section Analysis of declining accumulation rates with increasing geologic age

We next test a hypothesis of a uniform decline of average accumulation rate vs increasing age over the entire ocean. For this we examine the record of accumulation rate within a large number of individual sections. The accumulation rate for each segment of the age model for each section was expressed as the ratio of the mean rate for the whole section. The ratio value and midpoint age of each age model segment from each individual section were then compiled across all the sections in our study (see Figure 5). This had the effect of removing any between-section differences in section average accumulation rate, leaving only the temporal trend in change in relative accumulation rate vs age. We were looking for the expected signal, at least in the composite global dataset, of major declining average relative accumulation rate with increasing age, as had been reported by prior global summaries of deep-sea sections in the literature. Instead, we show that, despite substantial short term variance of individual age model segments within the individual sections, the composite of relative change vs age for the set of all sections did not show any detectable trend towards lower average accumulation rate with increasing age.

### 2.3  A Model of How Between Section Differences in Average Accumulation Rate Interact With Incomplete Recovery of Cenozoic Sections via Drilling to Create a Bias in Compiled Data

The near constant rates of relative accumulation found in our second sub-study, where we removed between site differences in accumulation rate, suggests that the signal of declining accumulation rate vs age as reported in prior studies must in some way be connected with these differences between sections in accumulation rate, and/or in how the data for different sections was composited in these prior studies. In the course of the above synthesis of relative accumulation vs age, it was apparent that a) most sections with high average accumulation rates ended in relatively young sediments, while sections reaching older sediments typically had low average accumulation rates; b) there was no obvious difference in the total depth (in meters below sea floor) reached by drilling between these different types of section; and c) only a few sections actually recovered the entire sedimentation record for a location (ie reached basement), or even recovered the entire Cenozoic time interval - most ended

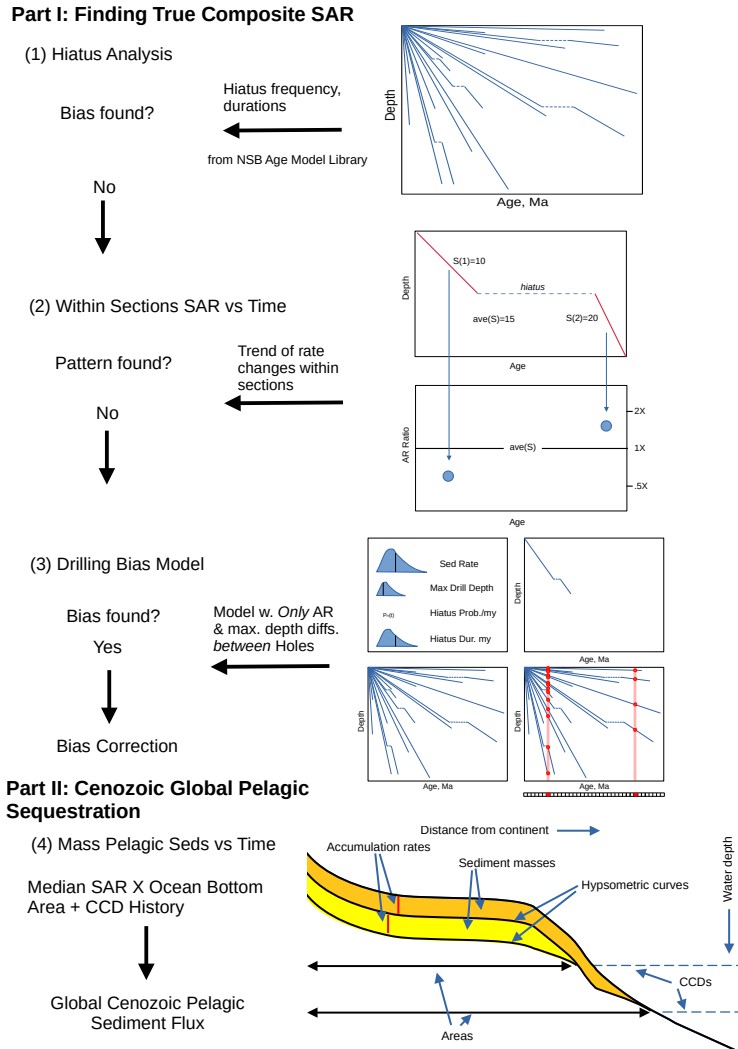

**Figure 2.** Workflow diagram showing sequence of analyses carried out in this study to determine true Cenozoic accumulation rate (AR) signal. 1) Calculated hiatus metrics in NSB database to see if common/large enough to, via Sadler effect, cause apparent rise in sedimentation rates over Cenozoic. 2) Relative AR change over time within individual sections, composited over all sections calculated. 3) Modeled bias of calculating binned AR by age from sections, each individually of constant rate, but differing in rate and maximum depth of recovery between sections. Model reproduces apparent rise in AR over Cenozoic. 4) Used raw AR data, adjusted for bias using curve from '3', Cenozoic ocean bottom area above CCD to calculate Cenozoic history of global AR of pelagic sediment. See main text for further description.

prior to base of the Cenozoic, constrained by limits imposed on total drilling depth from Leg priorities, bad weather, stuck drill bits etc. From this, it can be inferred that the age of recovered sediments in the global deep-sea drilling data is determined in part by constraints from the drilling process, not just primary sedimentary record; and, the only ways (barring the occasional

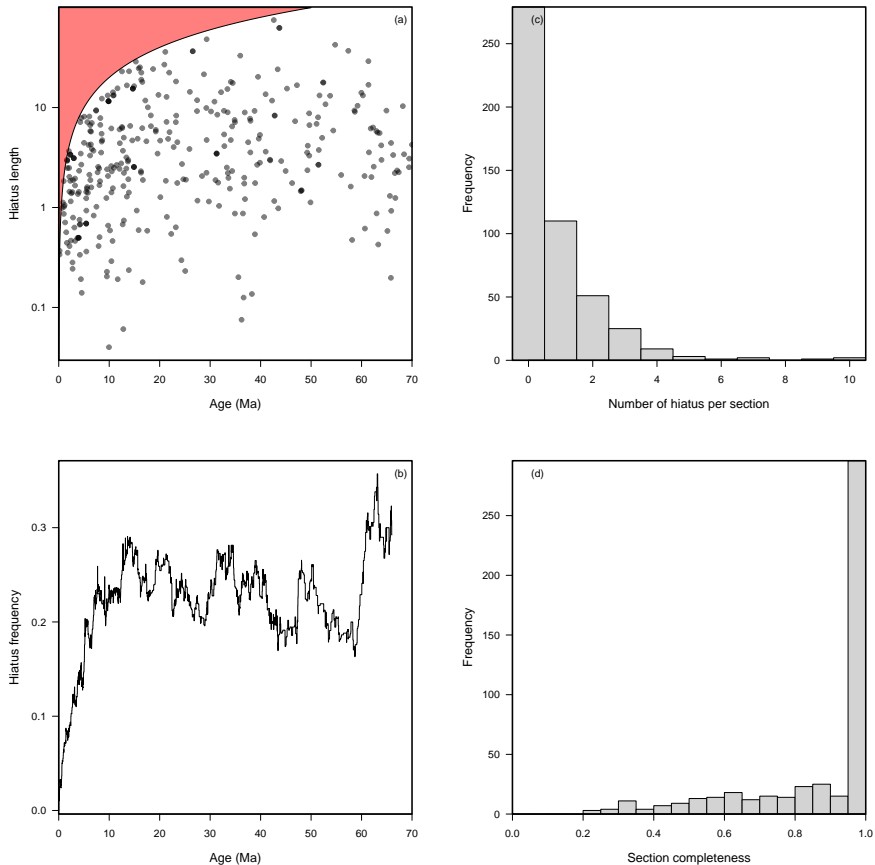

**Figure 3.** (a) Hiatus length through time (age of hiatus is its median age; in red is the area of the plot that thus can not be populated). (b) Hiatus frequency through time (i. e. what proportion of sections ranging through a given time interval do not recover that particular time interval). (c) Histogram of number of hiatus per section. (d) Histogram of section completeness (i. e. given a hole whose bottom core is at age x, what percentage of those x Ma were recovered).

major hiatus) to reach older time intervals is to have chosen (deliberately or by chance) to drill at a location with a low average accumulation rate, and/or to drill an unusually deep section. The complex history of ocean drilling, with its originally poor knowledge of local ocean sedimentation rates, and changing geologic time interval priority targets (Imbrie et al., 1987; Coffin et al., 2001, ; geologically young, high sedimentation rate sections; low sedimentation rate sections overlying geologically old target intervals, etc), has not however, over the course of the programs, systematically adjusted drilling depths on a global scale

to compensate for differing sedimentation rates. Thus, in global compilations of deep-sea drilling data, drilling depths can be treated as being random to local sedimentation rate. Further, if one were to compile (i.e. bin) accumulation rate data across sections by time interval, younger time interval bins would have a mix of both high and low average rate sections, and thus higher bin-average accumulation rates, than older interval bins, where only sections with low average within section rates can

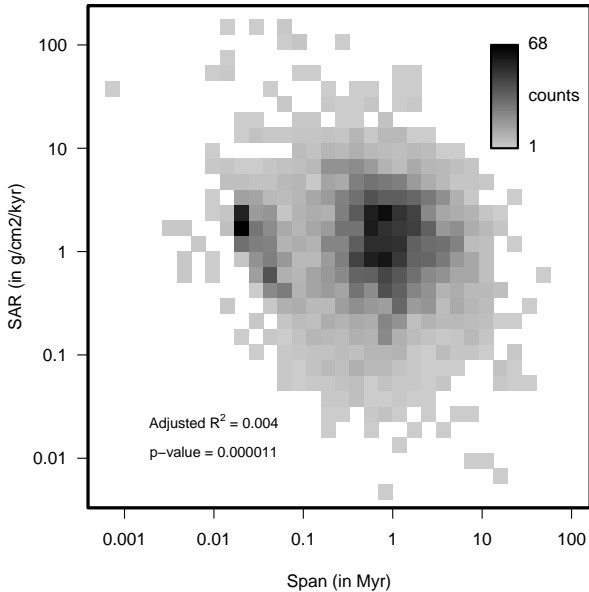

**Figure 4.** Density plot relationship between computed SAR and time span of the age model segment over which it was computed, following Sadler (1981).

normally be recovered. Davies and Worsley (1981, p. 170) were the first (and to our knowledge, only) workers to briefly note
that the drilling record is probably biased by preferential sampling of high sedimentation rates in younger sediments, although they did not investigate this issue, and simply concluded that this potential bias was not sufficient to affect their analyses.

We thus here explore the magnitude of this 'drilling bias' in a model, generating large numbers of model drill sections, and compiling (binning) the median accumulation rate across sections for 1 Myr time bins over the Cenozoic. Each model section was defined by just two main parameters: the accumulation rate for the section, and the total depth drilled by the section, each
190 picked from the observed distribution in real life data (see Figure 6). The model is set so that any individual section has a constant rate of accumulation vs time: the real median accumulation should thus be constant. Our model thus simulates how, with constant actual accumulation rate vs time at any location, the variation of average accumulation rates between sections, and random variation in drilling depths between sections, affects the composite (binned) accumulation rate across sections. The magnitude of this drilling bias turns out to be quite large, and in fact accounts for the entire increase in apparent binned
average accumulation rates across deep-sea drill sections over the Cenozoic. When the effect of this bias was substracted from the 'raw' across section binned data, the resulting 'drilling bias corrected' accumulation rate over the Cenozoic showed only a slight decline - and thus in broad accordance with the near constant rates seen in the second subanalysis of relative trends within sections vs time (see Figure 7). We also show how different ocean basins might have different Cenozoic accumulation rate histories (see Figure 8). These, when compiled together using a basin area adjustment matched as expected the global
curve. There were however some differences in individual basin histories.

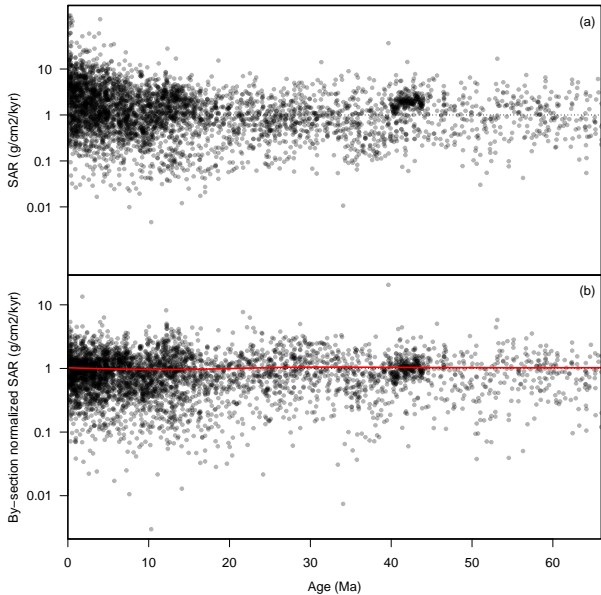

**Figure 5.** (a) SAR through time for each individual age model segment of a hole, compiled across all study holes' age models. (b) Same, but SAR of each segment normalized by mean SAR of its hole. Trend over time (LOESS; in red), showing the absence of a global pattern, on a hole by hole basis, toward increasing SAR over the Cenozoic.

## 2.4 A new pelagic sediment accumulation rate using the corrected SAR curve

Finally we apply our new, largely constant accumulation rates over the Cenozoic curve to the question of Cenozoic rates of global weathering. For this we examine only the dominant component of pelagic sediment output of weathering supplied dissolved nutrients - the calcium carbonate sequested in pelagic sediments by marine plankton. This was also done as relatively little information is available on the Cenozoic distribution of the other main component of weathering - biogenic silica in sediments; and in older intervals, how it has been affected by diagenesis to chert. To convert the accumulation rates for sections for each geologic time interval from our previous subanalysis into global accumulation rates per time interval of carbonate material, we calculate the area of the ocean accumulating carbonate for each time interval. The area of the oceans above a given depth (the CCD) was calculated from paleotopographic maps for each time interval of the Cenozoic and each ocean basin (see Figure 9).

## 3 Material and Methods

The Linear Sedimentation Rates (hereafter LSR) computed here derive from a much more comprehensive, and relatively unbiased selection of deep-sea drilling sites than has been used in prior studies. Specifically we use the age models for all 479 sites produced and revised, primarily by the authors over a period of nearly 30 years, for the Neptune (NSB) database (Lazarus,

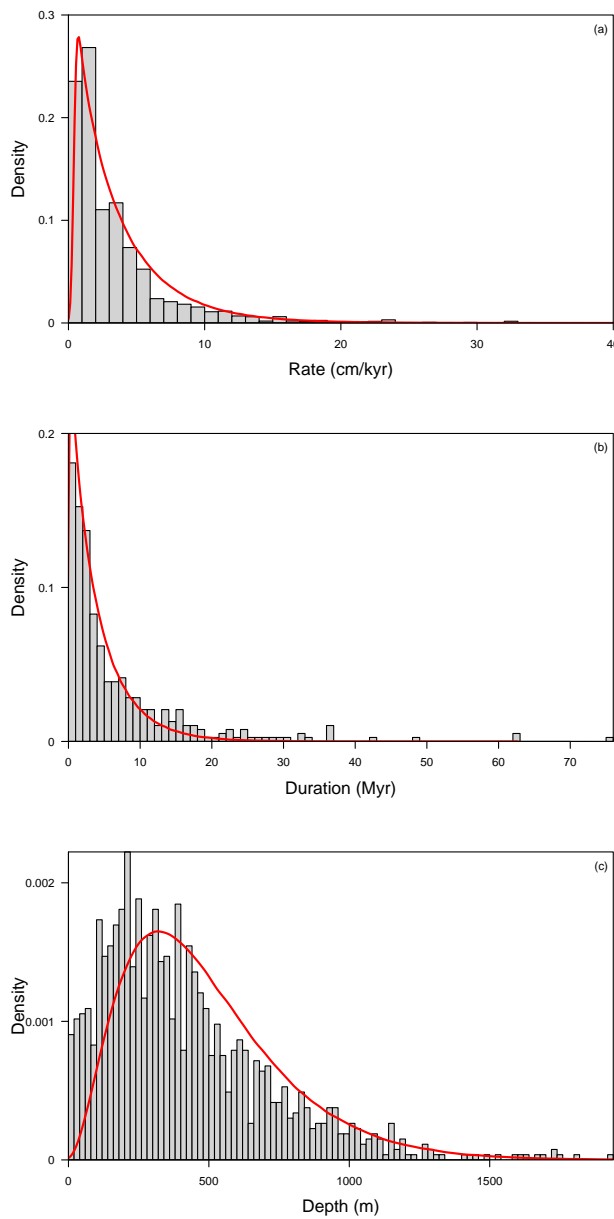

**Figure 6.** (a) Histogram of observed values of Pleistocene LSR in grey and density distribution of the modelled LSR (a translated Weibull distribution of shape $\kappa = 0.95$, scale $\lambda = 3.34$ and location $\theta = 0.4$) in red. (b) Histogram of observed hiatuses duration in grey and density of the modelled distribution (a Weibull distribution of shape $\kappa = 0.95$ and scale $\lambda = 4$). (c) Histogram of DSDP, ODP and IODP hole depths in grey and density of the modelled distribution (a Gamma distribution of shape $\kappa = 2.93$ and scale $\theta = 166.97$).

1992, 1994; Lazarus et al., 1995; Renaudie et al., 2020). Sites selected for inclusion in NSB meet the criteria of adequate recovery and sufficient information to construct a usable age model (meaning in practice at least adequate microfossil data) but otherwise are not selected for specific sedimentation criteria, or age intervals, and include mostly sites from the qualitatively better recovered ODP and IODP phases of the deep-sea drilling programs. LSR at each site was computed every 10 kyrs from 66 to 0 Ma; and global median LSR was computed, along with its interquartile range (IQR (Tukey, 1977)), based on all sites.

The moisture and density measurements for each site were extracted from the NGDC archive CD-ROM (National Geophysical Data Center, 2000, 2001), Janus (Mithal and Becker, 2006) and LIMS databases (lim, 2016). Densities for each site were used to transform each site LSR into Sediment Accumulation Rates (SAR), expressed in $g.cm^{-2}.kyr^{-1}$. These were compiled globally the same way as the global LSR was. As neither LSR or SAR distributions follow a normal distribution, medians are reported here rather than means; however for the purposes of direct comparison with the published literature which did use overwhelmingly means rather than medians, means are also reported and illustrated in the Supplementary Material.

Drilling depths, local SARs, hiatus frequencies and durations are all taken from the NSB database (Renaudie et al., 2020, 2023). Age models are generally accurate to ca 0.5 Myr, although some poor models are significantly less precise ($> 1$ Myr errors) and others (orbitally tuned sections) are much more precise (ca 40 kyr) (Renaudie et al., 2020; Smith et al., 2023).

Median LSR and SAR were computed for each ocean basin. The dataset comprises 220 holes for the Atlantic basin, 187 for the Pacific and 54 for the Indian Ocean. Using the paleobathymetric models of Straume et al. (2024), we computed the area of each of those basins through the Cenozoic. Using this, we recalculated the global SAR curve as an area-weighted composite of the regional curves, to test for geographical biases.

Given the area covered in red clays is not representative of biogenic pelagic sedimentation, being mostly eroded but un-weathered terrigenous dust sized material (Glasby, 1991), we also computed the 'red clay'-free basin areas by tallying the area whose paleodepth was above the carbon compensation depth (hereafter CCD) for each basin, using both the paleobathymetric models of Scotese and Wright (2018) (accessed with the R package chronosphere; Kocsis and Raja, 2020) and Straume et al. (2024). The CCD used for the Pacific Ocean is that of Pälike et al. (2012), for the North and South Atlantic Dutkiewicz and Müller (2022) and for the Indian Ocean Van Andel (1975). The 'red clay'-free composite SAR was also calculated.

Lastly, our estimate of the total global sediment flux history (in $Pg.yr^{-1}$) is calculated over the last 55 million years only, as the previous 11 million years are not covered by all the published CCD estimates.

## 4 Results

### 4.1 Analyses of hiatus abundances and sedimentation.

The relation of SAR to time span of measurement is very weak, with an adjusted $R^2$ of only 0.004 (Figure 4). Unrecognized hiatuses in longer age model segments (e.g. in poorer age models) thus do not have a significant effect on SAR. Nor do age model segment spans increase with geologic age (Figure 3). Age model hiatuses (>0.5 my) in NSB deep-sea sediment sections are relatively rare (median 0, mean 0.93 hiatuses/section; Figure 3). Most are of fairly short duration (median 2.98 Myr), though there is an extended tail of rare, longer duration hiatuses, thus a mean duration of 6.47 Myr (Figure 3). Overall, the section

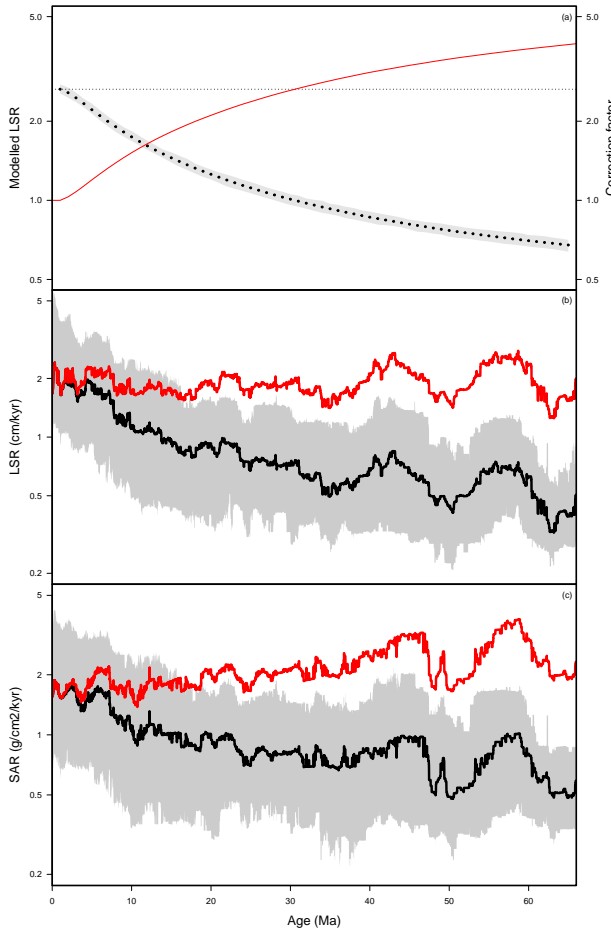

**Figure 7.** (a) Bias towards lower apparent sedimentation rate in older time intervals in drilled sections. Results from a model where there is no actual change in local accumulation rate over time, rates however vary geographically. These sediments are then sampled by drilling sites with a randomly chosen drilling depth, drawn from the distribution of drilling depths of all deep-sea drilling program sites. The *apparent* sedimentation rate over time compiled from the drilled sections is not constant but declines with increasing geological age, in part due to sediment compaction, but primarily by biased sampling of the older record. In black, median LSR averaged over 1000 runs (in grey, full extent of the various runs), in dotted actual median LSR inputted in the model, and in red, correction factor derived from it.

(b) and (c) Linear Sedimentation Rates (c; expressed in $cm.kyr^{-1}$) and Sediment Accumulation Rates (d; expressed in $g.cm^{-2}.kyr^{-1}$). Black: distribution as observed in our global dataset (solid line: median; grey shading: interquartile range); Red: median sedimentation and accumulation rates corrected for drilling bias using the result of the model). Rates are given on a log-scale.

age models are on average >85% complete (fraction of time represented by sedimentation) (Figure 3).Hiatus durations do not become longer with increasing age, other than the automatic effect of truncation in maximum possible duration in younger sediments (Figure 3). Hiatuses do overall become more frequent with increasing age (Figure 8) but the pattern is divided into

three distinct intervals. Hiatuses are at near zero frequency in the youngest time interval but increase more or less linearly in frequency until approximately 10 my, after which, despite some oscillations, they show a very slight decline throughout most of the Cenozoic, from about $24 \pm 4\%$ in the mid Miocene to $22 \pm 4\%$ in the Eocene. They again rise in frequency in the mid Paleocene to a higher plateau of ca 30% in the early Paleocene. (Figure 3).

## 4.2 By.section analyses.

SAR for individual age model segments in the NSB library vary dramatically ($< 0.5$ to $> 30 cm.kyr^{-1}$ range; interquartile ratio 4.43; Figure 5). By-section normalized rates vary more modestly (interquartile ratio of 2.38); much of the SAR variation thus exists between sections, varying only moderately within a section. Crucially, there is no trend in by-section SAR over geologic time when compiled for the entire dataset (Figure 5): the slope of a linear regression is insignicantly different from zero, and the $R^2$ is also essentially zero ($1.277 \times 10^{-5}$). Variations in rate within sections thus are modest and essentially random vs time.

## 4.3 Sampling bias by drilling.

Using the sedimentation rates, hiatus durations and frequency calculated from the NSB database, our model of the effect of drilling bias on apparent sedimentation rates in binned data gave a consistent result, with only little variation between individual runs, each with 1000 simulated holes. The binned rates of sedimentation per time interval over the Cenozoic decline substantially due to drilling bias with increasing geologic age (Figure 7). The magnitude of the decline is very similar however, with mean sedimentation rates of ca $3.9 cm.kyr^{-1}$ in the youngest time bin (0-1 Ma), declining to $< 0.7 cm.kyr^{-1}$ in the oldest (64-65 Ma), or approximately a 6-fold decrease in apparent rate.

## 4.4 Bias-corrected global Cenozoic rates of sedimentation and accumulation

Our more comprehensive, NSB-based global compilation of LSR and SAR (Figure 7) shows that the apparent (i.e. uncorrected) increase through the Cenozoic is less drastic and more uniform than reported in previous works (Hay et al., 1988; Westacott et al., 2021). Corrected using our model output for drilling bias, the actual LSR and SAR, far from increasing throughout the Cenozoic, seem to have decreased slightly (Figure 7), by a factor of 2 (from an estimated $4 g.cm^{-2}.kyr^{-1}$ in the Paleocene to ca. 2 in the Pleistocene).

This global pattern hides regional disparities (Figure 8). The Pacific Ocean had a relatively stable sedimentation history, with a near constant early Paleogene SAR that decreased abruptly starting from ca. 41 to 35 Ma, before increasing rapidly shortly before the Eocene-Oligocene boundary to its Cenozoic maxima, followed by a somewhat variable but gradually declining rate to the Recent. By contrast, the Atlantic Ocean sedimentation history shows a more monotonous decrease throughout the Cenozoic with two more abrupt decreases: one near the PETM and another one in the Early Miocene. The Indian Ocean shows a decrease in SAR in the Early Eocene and a short term increase from ca. 40 to 36 Ma.

Despite these disparities, the area-corrected global SAR curve (Figure 8) is remarkably close to the global median, and shares the same characteristics. Apart from its main decreasing trend, it shows a pronounced decrease at ca. 55 Ma, ending at ca. 49 Ma; and a slower decrease from 42 to 35 Ma, terminating in a fairly sharp increase from 35 to 33 Ma. The same area-corrected global SAR curve once corrected to remove the red-clays area differs from the other two curves in that the overall level is lower in the Paleogene thus reducing the overall Cenozoic decrease. The pattern discussed above still holds true though.

The global pelagic biogenic sediment flux (Figure 9), correcting for varying area undergoing pelagic biogenic sedimentation due to changing CCD, shows a near stationary, though slightly decreasing (with a mean around $2.5 Pg.yr^{-1}$), Eocene sediment flux, a sharp increase in the latest Eocene from ca. 35.1 Ma to 33.9 Ma, and a net stationary Oligocene to Recent sediment flux (though marked by wide variations, oscillating from ca. 3 to ca. $6 Pg.yr^{-1}$, around a mean of 4 to 4.5, depending on the chosen paleobathymetric model). Although these values are significantly lower than current estimates of Quaternary land-to-sea sediment fluxes (McLennan, 1993; Walling and Webb, 1996; Syvitski et al., 2022), our dataset excludes coastal areas, which are dominated by eroded terrigenous, not chemical weathering produced sediments.

Lastly, we tested the sensitivity of our result to the quality of the age models used by running the analyses excluding holes with poor quality age models. The curve is very close to that including all age models (Supplementary Figure 3), suggesting that our result is unlikely to have been affected by age errors in the data.

## 5  Discussion

### 5.1  Hiatus analyses

Prior studies of hiatus frequency and duration (Moore Jr and Heath, 1977; Moore Jr et al., 1978; Dutkiewicz and Müller, 2022) do show some similarities to our results, in having a prominent decline in frequency in the last few my, and an increase below ca 60 Myr. The decline in the youngest time intervals may in part be an edge effect, due to the truncation of possible hiatus lengths (the red shaded interval in Figure 3a), and by the difficulty of detecting short hiatuses using the age model approach. This is because too little section is available above a possible hiatus depth to clearly define a line of correlation segment that does not intercept the line of correlation segment below the possible hiatus. Drilling disturbance and non-recovery of the upper meters of sediment are also common in deep-sea drilled sections and these mechanical disturbances of the uppermost sediment record may also blur the record of any short hiatuses in the section. The decline in hiatuses in younger sediments may however also reflect a real phenomenon, Dutkiewicz and Müller (2022) interpreted the late Neogene decline in hiatus frequency as a primary paleoceanographic phenomenon, attributed to declining intensity of abyssal circulation. We suggest that the decline in more recent sediments represents an erosional process, in that hiatuses may not only develop by intervals of non deposition, but also post depositionally by slumping and bottom current erosion in the upper part of the sediment column. These sedimenents are not only more exposed to erosional activity but also in the upper ca 100-200 meters softer and uncompacted, and thus more easily eroded. The effects of such processes on the youngest sediment layers would be cumulative, until deeper burial brings the process to a halt, and would produce the pattern observed. This can be viewed as a one specific parameterization of the

more general model of near continuous, if varying intensity post depositional sediment loss presented by Moore Jr and Heath (1977).

We do not have any explanation for the distinctly higher rate of hiatus development below ca 60 Myr. This interval is directly above the mass extinction event of the K/Pg boundary, and we speculate that it may in some way be related to ocean conditions during the recovery interval, including ecologic effects on the utility of biostratigraphic zonations (see also below), or comparatively widespread regions of hiatus development or areas on non-deposition in low productivity post-impact oceans (D'Hondt, 2005). Dutkiewicz and Müller (2022) alternatively suggested increased erosion by bottom water circulation as a

possible cause. However, as also noted by Dutkiewicz and Müller (2022b) the number of sections available for study in this time interval is very low and the high hiatus frequency could also simply be due to insufficient age model data.

Early studies also found much higher frequencies of hiatuses (i.e. 50-80% Moore Jr and Heath, 1977) than seen in either our study, or the recent study of Dutkiewicz and Müller (2022). There are several possible reasons for this. Early drilling was based on very poor knowledge of deep-sea sediment content, while later drilling used improved information, e.g. high resolution

seismics, to target better sections. Thus the compilation used in our study is likely to have some bias towards more complete sections. Hiatuses however were determined in early work not by the construction of detailed age models but by counting any 'missing' biostratigraphic zones as hiatuses. This approach assumes that zones will always be identifiable in continuously deposited sections. As the field of Cenozoic marine biostratigraphy has matured it has become abundantly clear that many zones are not always identifiable in sections due to other reasons, such as the rarity or complete absence of one or more zonal

marker species due to biogeographic/ecologic restriction, poor preservation, or difficulties in taxonomic identification. Much work in recent decades has thus gone into finding alternate zonal markers or defining more regional zonations to combat these limitations (Raffi et al., 2006; Scherer et al., 2007; Wade et al., 2011; Lazarus et al., 2020). Thus many of the absent zones used in earlier studies as evidence for hiatuses are likely to be simply imperfect biostratigraphy.

Regardless of the above details, our results clearly show that hiatuses are far too infrequent, and show far too little change in

frequency or size over most of the Cenozoic, to have had any significant impact on apparent sedimentation rates. The change signal therefore must be held in the dominant (sedimenting) part of the sections.

## 5.2    Drilling bias and its effect on data compilations of sedimentation rate

It is clear that the bias effect arises only because drilling depths are usually too short to recover the entire sedimentary section: otherwise, high sedimentation rates would also be recovered without bias even in older sediments, despite the thicker section

that was drilled. The magnitude of the bias when this is not so largely reflects the range of sedimentation rates between hole locations, from mean modern rates in the Recent to the values near the lowest modern rates for the oldest recovered sediments (Figure 5). Lastly, the effect of including hiatuses in the model is simply to reduce the magnitude of bias vs increasing age, as drilling is effectively extended in depth by the magnitude of the hiatuses. The effect however remains, even if the full magnitude is not reached until earlier in geologic time.

Although we have not explored the implications of bias beyond that of drilling pelagic sediments, it is to be expected that a similar effect exists in other types of drilled geologic sections; and even to some degree ordinary land sections, insofar as

exposure of older time intervals via erosion or tectonic movements is at least in some degree dependent on the sedimentation rates and thus thickness of the overlying younger sedimentary section.

## 5.3 A new estimate of Cenozoic weathering history

The drilling bias analysis shows that the signal of (ca 6X) increase over the Cenozoic is in fact an artifact, due to the tendency of drilling to only sample older sediments where local rates of sedimentation are low. Once corrected for this bias global rates of pelagic sediment accumulation show a much more stable pattern over time, albeit also marked by a step-like increase near the Eocene-Oligocene boundary, and several short term fluctuations.

Global mean deep-sea pelagic sediment accumulation rates have decreased slightly during the Cenozoic. This contrasts
sharply with all previous estimates (Davies et al., 1977; Worsley and Davies, 1979; Hay et al., 1988; Westacott et al., 2021) which described an increasing trend, in particular in the Neogene. Our result also differs, if less strongly, from the estimate of constant accumulation between ca 5 and 65 Ma, and higher rates in the last 5 Ma, from seismic data based global sediment thickness maps (Olson et al., 2016). The latter study is not however directly comparable as it includes both terrigenous and biogenic pelagic sediments. Ocean sequestration processes thus have changed over Cenozoic. Despite increasing temperature
latitudinal gradients and inferred stronger circulation forcing, both which may have increased ocean turnover, nutrient transfer to surface waters and thus productivity, local sequestration rates have declined. Possible causes are beyond the scope of this paper but could include lower local primary or export productivity, possibly due to lower nutrient concentrations in water, reduced transfer of nutrient waters to surface layers (despite stronger global wind stress) due to increased vertical thermal gradients and stratification, or reduced efficiency of sequestration due to increases in e.g. deep water dissolution. Colder deep
oceans since the Oligocene though tend to reduce chemical process rates including dissolution rates – $C_{org}$ (Burdige, 2011); silica (Westacott et al., 2021); carbonates (Morse and Arvidson, 2002) –, thus a reduced efficiency of sequestration seems unlikely.

Adjusting our accumulation rate histories for the changing area of oceans accumulating biogenic sediment however yields a more stationary trend for global biogenic sediment flux to the deep-sea, with the difference that the Oligocene to Recent mean
is significantly higher than the pre-EOT one. This shift generally supports prior studies that have suggested that changes due to continental glaciation of Antarctica near the Eocene-Oligocene boundary would have altered long term pelagic biogenic sediment accumulation rates, either by loss of shallow sea carbonate accumulation area (e. g. Kump and Arthur, 1997; van der Ploeg et al., 2019; Komar and Zeebe, 2021), higher values of 'weatherability' (Kump and Arthur, 1997).

We do not see any trend towards higher rates of pelagic accumulation over the later Cenozoic, and in particular over the last
10-15 Myr. This supports studies that have suggested steady state over the late Neogene (e. g. Willenbring and von Blancken-burg, 2010, , see Figure 10) and stands in contrast to studies that have suggested substantial increases over the same time period (van der Ploeg et al., 2019; Caves Rugenstein et al., 2019; Westacott et al., 2021). van der Ploeg et al. (2019) estimated the fraction of carbonate deposited in shallow environments as the difference between deep-sea burial and estimated weathering flux (calculated using a model driven by Sr and Os isotope variations and Cenozoic CCD history). They found that, at least
since the Oligocene, shallow carbonate burial was only a fraction of that of the deep-sea (generally 4 times higher in pelagic

sediments). Kump and Arthur (1997) study showed even stronger disparities, with Paleocene-Eocene pelagic carbonate accumulation rates twice those of shallow environments, increasing to an 8-fold difference in the Oligocene-Recent. These low estimates of Cenozoic shallow vs deep water carbonate accumulation suggest that the potential impact of changes in shallow settings is, other than near the Eocene-Oligocene boundary, rather limited , and thus the differences between our results and those that call for substantial increase in Neogene accumulation rates cannot be due to changes in shallow-deep partitioning of carbonate burial. The above models' results also contrast with the estimate of Milliman (1993). One possible reason may be the difficulty in scaling Milliman (1993) short time scale measurements of high rates of accumulation in many geographically limited shallow settings to longer time scales due to Sadler type effects. These would, as shown in our current study, primarily affect shallow environments, and would thus result in much lower longer time scale rates of accumulation in shallow water vs pelagic settings.

In general, our results most closely resemble those of Kump and Arthur (1997), although the very low temporal resolution of their study prevents a more detailed comparison. Lastly, our curve shows substantial short term changes in flux on time scales of 5-10 Myr, particularly in the Oligocene to Recent. Although these could reflect dynamics of system feedbacks, we have not carried out a detailed analysis to see if these fluctuations might not be due to artifacts and limitations of our data and methods, and we therefore do not interpret them further.

## 5.4 Limitations of our study

Uncertainties in our results due to poor age model quality are probably minor given the large number of sites used, and experience of high stability in biodiversity compilations from the same suite of sections using successive iterations of age model development (Lazarus et al., 2020)(see also Supplementary Figure 3). Inconsistent coverage by age interval and geographic region may affect short intervals in our results but are unlikely to significantly affect the basic trends documented. he one possible exception to this is that subduction has removed a substantial portion of the earlier record, and the subducted regions, being mostly relatively near ocean margins, may have preferentially removed near coastal, and on average higher sedimentation sections associated with coastal upwelling regimes. This would have the effect of reducing the potential bias calculated by our model. Lastly, CCD trends in other ocean basins, while not identical (Dutkiewicz and Müller, 2022), largely follow the trend for Pacific CCD, and given the relative size of basins, differences are not likely to substantially alter the main trend in global flux.

## 5.5 Broader implications of drilling bias for paleoceanography and marine micropaleontology

Although paleoceanographers have long known that older sediments are less commonly recovered than younger ones (Lisitzin, 1996) they have, to our knowledge, largely accepted the recovered record as an unbiased, if ever sparser representation of (older) ocean sediments. Here we show that the Cenozoic deep-sea drilling ocean record has a substantial systematic bias. Compilations of paleoceanographic data by time interval need to consider that the older intervals have a substantial over-representation of low sedimentation rate conditions, and an increasingly poor representation of high sedimentation rate environments. This bias may affect a variety of geochemical and micropaleontological synoptic studies that attempt to compare how ocean con-

ditions have changed over the Cenozoic. More broadly, this sampling bias may affect our understanding of any past ocean parameter that varies with either sedimentation rate itself, or with the ocean conditions correlated to sedimentation rate, such as surface water productivity. As these parameters influence a very broad range of ocean processes and chemical cycles (Berger et al., 1989; Morel and Price, 2003; Sarmiento and Gruber, 2006; Hüneke and Henrich, 2011), the discovery of this bias thus will have broad significance, potentially requiring significant reassessment of published literature (Hay et al., 1988; Westacott et al., 2021), as well as affecting the design of many future paleoceanographic and micropaleontologic studies.

## 6 Conclusions

The record of pelagic sedimentation recovered by deep-sea drilling is systematically and substantially biased. Older time intervals are increasingly represented only by lower accumulation rate sections. This is due to limited total drilling depths and the consequence that reaching older ages happens mostly when drilling in low sedimentation rate sections. This bias will affect any study that compares compilations of data across deep-sea sections over time, where the parameter being compiled is sensitive to local accumulation rates. This includes a broad range of paleoceanographic and microfossil ecologic proxies, and others. The conclusions of studies based on such data that have already been published may in some cases need to be revisited, and the bias problem will need to be addressed in designing similar future research. The global average accumulation rate in local deep-sea pelagic sediment sections has declined slightly over the Cenozoic. As a greater percentage of ocean basins have accumulated sediment over the Cenozoic via deepening of the CCD, the global flux of biogenic sediment has increased, however mostly by a single stepwise increase near the Eocene-Oligocene boundary. The lack of any significant trend in global pelagic accumulation rates since the Oligocene supports arguments for similarly little net change in long term Oligocene-Recent global weathering rates.

*Code and data availability.* The code for the model, as well as its ouput and the data presented here are all available in the Supplementary Materials. Age models are also given in the Supplementary materials while the underlying stratigraphic data are available from the Neptune (NSB) database (Renaudie et al., 2020, 2023). Additional code used to run the analysis can be found on a Zenodo repository (Renaudie and Lazarus, 2024).

*Author contributions.* Both authors co-developed the age model library in NSB; jointly designed the study and co-wrote the manuscript. DBLwas lead author for workflow sections 2.2 (by section analyses) and 2.3 (drilling bias model); JR for everything else: sections 2.1 (hiatuses, Sadler effect) and 2.4 (global SAR, consequences for weathering history).

*Competing interests.* The authors declare no conflict of interest.

*Acknowledgements.* This study was funded by the Federal Ministry of Education and Research (BMBF) under the "Make our Planet Great Again – German Research Initiative", grant number 57429681, implemented by the German Academic Exchange Service (DAAD).

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

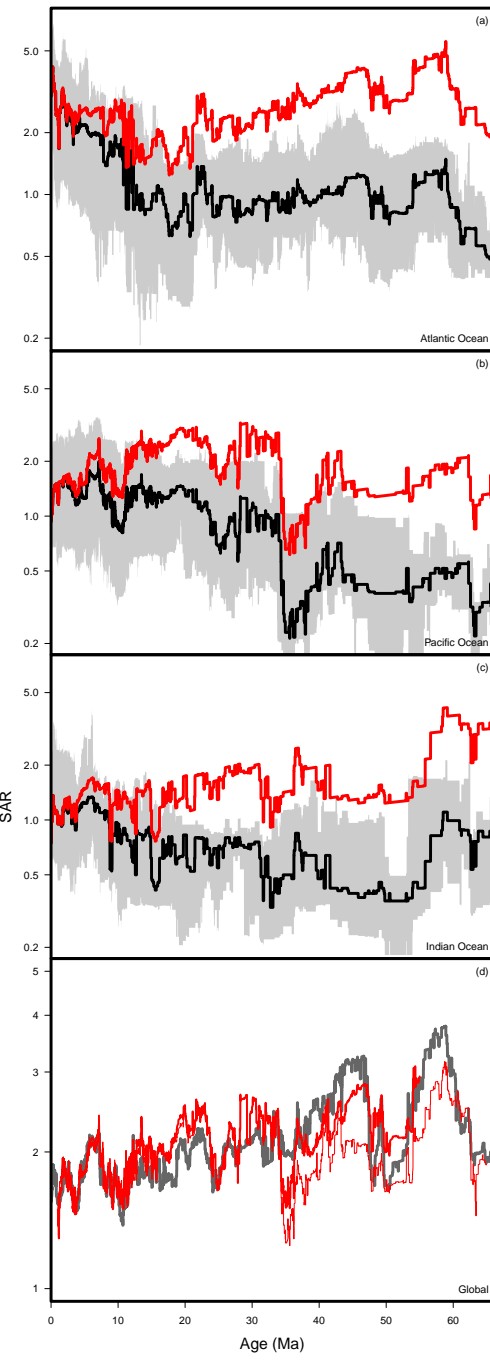

**Figure 8.** Sediment Accumulation Rates (expressed in $g.cm^{-2}.kyr^{-1}$) distribution observed in each ocean basin (black: median; grey: interquartile range; red: median corrected for drilling bias using the result of the model), shown here on a log-scale. Bottom panel (d) show the weighted global composite based on each basin corrected median value (in bold red: full marine area; in dotted red: 'red-clay'-free) compared to the corrected, but not area-weighted, global median SAR shown in Figure 7.

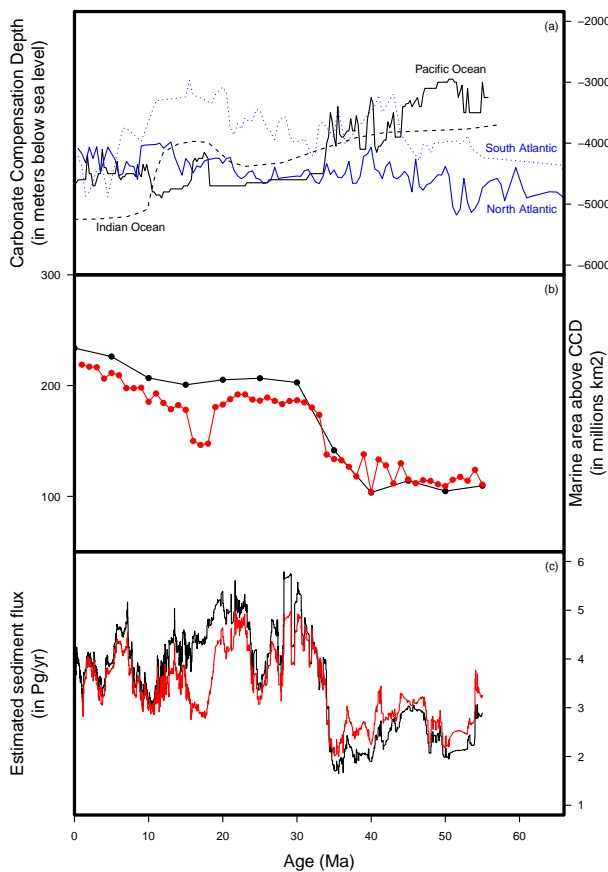

**Figure 9.** (a) CCD history for each basin, in meters below sea level (Van Andel, 1975; Pälike et al., 2012; Dutkiewicz and Müller, 2022).(b) Estimate of changes in the global depositional area of pelagic sediments, i. e. area above the Carbonate Compensation Depth (CCD), expressed in millions of $km^2$. Calculated from the CCD above and the paleobathymetric maps of Scotese & Wright (Scotese and Wright, 2018; Kocsis and Raja, 2020) in black and of Straume et al. (2024) in red. (c) Estimated flux of sediments throughout the Cenozoic, expressed in $Pg.yr^{-1}$, based on the median SAR of each basins (Figure 8) and the changes in depositional area of each basin.

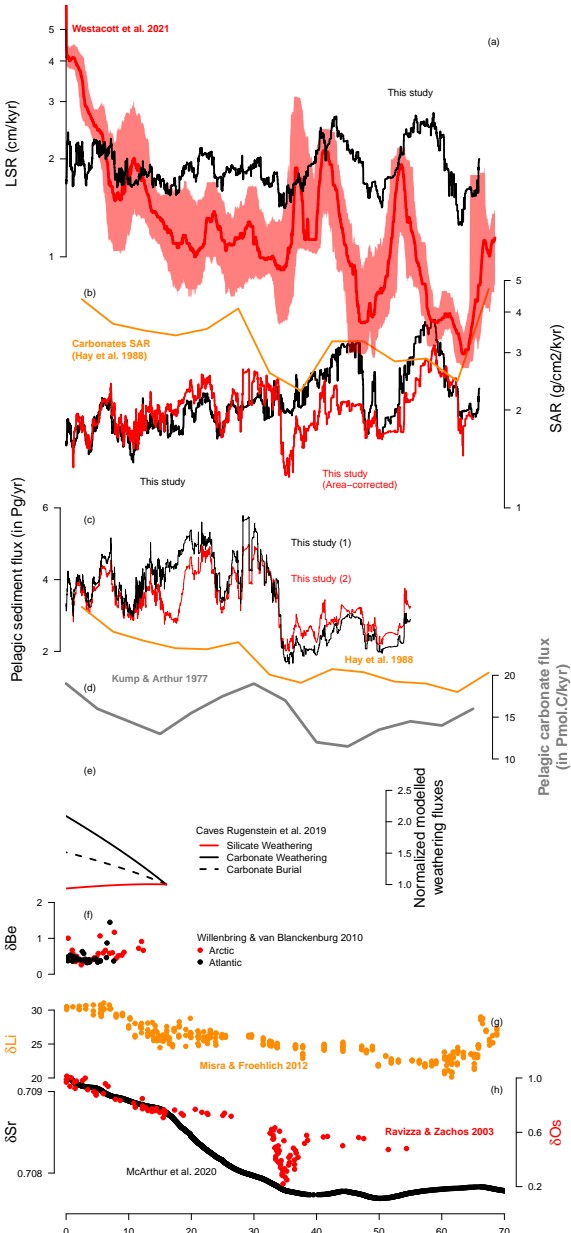

**Figure 10.** Comparison between various proxies and model outputs for Cenozoic weathering, and reconstructed sediment accumulation rates or fluxes from the literature and this study. (a) LSR reconstructed here (in black) and in Westacott et al. (2021) (in red, with confidence interval). (b) SAR reconstructed in this study (in black from Figure 7 and in red, area-corrected, from Figure 8) and Carbonates SAR from Hay et al. (1988) (in orange). (c) Pelagic sediment flux reconstructed here (from Figure 9, in black using Scotese & Wright paleobathymetry and in red using Straume et al. paleobathymetry) and from Hay et al. (1988) (in orange).(d) Pelagic carbonate flux reconstructed in Kump and Arthur (1997). (e) Modelled carbonate (in black) and silicate (in red) weathering fluxes and carbonate (dashed line) burial flux from Caves Rugenstein et al. (2019). (f-h) Weathering proxies: beryllium isotopes (f) from Willenbring and von Blanckenburg (2010), lithium isotopes (g) from Misra and Froelich (2012), osmium isotopes (h, in red) from Ravizza and Zachos (2003) and strontium isotopes (h, in black) from McArthur et al. (2020).