# Peer review of "Cenozoic pelagic accumulation rates and biased sampling of the deep-sea record"

_EGUsphere, 2023_

## Author Response (AR1)

We are hereby resubmitting our manuscript 'Cenozoic pelagic accumulation rates and biased sampling of the deep sea record' for consideration for *Biogeosciences*. We modified substantially the manuscript to accomodate for the reviewers requests. In particular, the main points of change are the following:

- We added a section titled 'Workflow' before the Materials and Methods, along with a new figure illustrating the MS workflow, to explain more clearly the logical rationale of the study.

- We added significant reviews of past literature on sedimentation rates as well as on Cenozoic weathering rates, in order to make the discussion related to this matters clearer to the readers. Similarly we added two new figures to illustrate and compare those past studies to our new time series. Due primarily to the more extensive review of prior work and extended Discussion section, the length of the the ms text (including captions but not abstract, references etc) has approximately doubled, from 3.7 to 7.8K words. We have also added over 2 dozen additional references.

- We redid the computation translating the global median sediment accumulation rate into estimated global sediment flux to account, as both reviewers suggested, for differences in CCD history between basins. One reviewer also suggested using the global CCD estimate of Boss and Wilkinson 1991: we also produced an estimate based on that CCD curve but only added it to the supplementary material as the methodology used to obtain that largely featureless CCD estimate was not clear to us.

In addition to submitting the corrected manuscript and a version of it with changes highlighted, please find detailed points by points answers to the reviewers queries hereafter.

The authors,
Johan Renaudie and David Lazarus.

**Reviewer 1**

This is an interesting and useful paper that describes a new approach to understanding the extent to which pelagic sediment accumulation rates in the global ocean have changed through the Cenozoic, and to what extent these observed changes are the result of drilling bias. The paper is novel and will be of interest to a wide readership using scientific ocean drilling data. The manuscript should be published after moderate revisions, that include clarification of the methodology, reconsideration of the CCD model used, better illustration of depositional area computations, and inclusion of new figures (locality map, methodology flow chart, seawater Sr curve). The methodology used in the analysis is very difficult to follow, and needs to be explained/visualised better.

We added a Workflow section to better explain the methodology, and added a flow chart. We modified the CCD mode used and provided depositional area maps in a repository. Reconstructed Cenozoic Sr curve is shown along with other geochemical proxies in a figure comparing them with our new timeseries.

The statement "The accumulation rate of pelagic marine biogenic sediments are thus a measure of weathering history" (line 3) is incorrect and should be reworded. Perhaps replace "thus a measure" with "an indication", bearing in mind that not all continental weathering products end up on the seafloor (e.g., dissolution of carbonate) and not all weathering products come from continents (e.g., seafloor weathering).

Corrected.

In the abstract the authors state that "When accumulation area however is adjusted for changes in available deposition area, global weathering is shown to have nearly doubled at the Eocene-Oligocene boundary." It would be useful to see maps of these time-dependent deposition areas, and a graph showing the time-dependence of these areas in individual ocean basins in the manuscript or supplement. The statement also assumes a straight forward correlation between deep-sea sedimentation and continental weathering, which is an oversimplification. The authors also need to explain why this would have happened at the EOT.

Other comments about the abstract: L1: capitalize Earth (and elsewhere in the manuscript) L2: $pCO_2$ is written incorrectly L3: change "are" to "is" L6: change "deep sea drilling" to "deep-sea drilling" (do this elsewhere and for other compound adjectives in the manuscript)

Done.

> Methodology The hypothesis is that the older the sediments recovered are, the more likely they are to be represented only by low sedimentation rate sites, as penetration to older aged sediments is more likely when the local sedimentation rate is low. I have no doubt that this effect exists, but it is questionable whether this is the only effect driving the observation that sedimentation rates decrease with age.

How preponderant this effect is is precisely what we tested using our model.

> The bias correction is based on this: "The inverse of the resulting modelled LSR vs geologic age curve is used as a correction factor on the NSB-based LSR and SAR compilation." I don't understand why the correction for this effect is based on LSRs which are not corrected for compaction (unless I understand it incorrectly, and the modelled LSR with age includes a compaction correction). Compaction clearly plays an important role in decreasing apparent LSRs with increasing age, so the compaction effect needs to removed first. This automatically happens when calculating sediment accumulation rates (SARs), so isn't it logical to compute the bias-correction from SARs, as opposed to LSRs which themselves suffer from a bias from compaction, which in turn could bias a bias correction?

SAR are computed by multiplying LSR with the actual observed density of the sediment (hence correcting for compaction): if we apply a compaction correction beforehand, we would overcorrect for compaction.

> Next, the authors are following the assumption that the long-term LSR change with age is due to a drilling bias, and this is implemented by applying the inverse of this relationship as bias correction. The problem that I see is that if your assumption is incorrect, in the sense that there are long-term reductions in sedimentation rate that are not in fact due to drilling bias, then the "corrected" accumulation rates will be biased themselves.

We are not assuming this: we model this bias, measure it and account for it. There are probably other unknown biases, but until we figure them out, measure them and account for them we can not correct for them. The 'corrected' accumulation rates we present are corrected for the biases we were able to identify and measure.

I see some potential ways for further testing. One optional thing that could be done is to exclude all sites that were drilled in very high productivity areas, i.e., sites that were deliberately focused on Neogene very high sedimentation rate areas, including most sites in the equatorial Pacific, and sites from coastal upwelling regions (e.g., Benguela current sites offshore Africa etc). Likewise, regions of sediment focusing (contourite drifts) are also anomalous and would need to be excluded. What do the rest of the sites look like in terms of LSR with age? I realize that this would be a fair amount of work so perhaps the authors can assess if this is worthwhile doing.

This would indeed be interesting but goes beyond the scope of this paper.

Figure S2 shows that the accumulation rate data uncorrected for drilling bias (black curves) are quite different between the ocean basins. In this context it would be of interest to see the uncorrected LSR data subdivided into ocean basins as well, to get a sense of what the curves look like before the correction is applied (keeping in mind that the correction is a model that might perhaps be incorrect). For instance, in the Atlantic Ocean, the "raw" SARs are essentially constant at times before 15 Ma (long-term trend) while the Pacific Ocean shows a long-term decrease in SARs with increasing age, the Indian Ocean somewhere in between, and the Southern Ocean shows ups and downs entirely different from the other ocean basins. With so much inter-basin variability, I start wondering how meaningful a global correction for sedimentation rate with age is, and how valid the premise of the paper is (but again, this is difficult to judge without seeing the uncorrected data, which should be shown).

Hence the reason why we present a per-basin analysis. The uncorrected data is shown on all those figures. We just chose to only show SAR and not LSR in the later figures as this is the one that is relevant to the discussion (in particular since it is compaction-corrected by definition).

In Figs 7 and S1, are the linear sedimentation rates shown corrected for compaction? This is not clear from the captions. In the text, please provide an estimate of how much the sedimentation rate would have been affected by compaction.Fig. 6 shows corrected linear sedimentation rates, and corrected SARs. The figure should show both uncorrected and corrected rates, as well as the correction function, to make this more transparent.

The correction factor was added to the figure showing the result of the model. Given the nature of Figure 6 (testing the effect of age model quality on the final result) we do not think adding uncorrected rates would be helpful. LSR

is never given corrected for compaction, as this is what SAR basically are and thus would be redundant.

> Lastly, the entire workflow is so complex that a casual reader of this paper currently would have no hope to understand what was actually done. It is quite possible that some of my comments above merely reflect that I didn't fully understand the methodology. This can be fixed by a flow chart that covers each step in the analysis, to make the paper clearer and more comprehensible to a general audience.

We added such a flowchart as Figure 1.

> Deposition area computation A simple global median curve based on 'red clay'-free oceans (i. e., 74.2% of the Atlantic, 51% of the Pacific and 74.7% of the Indian Ocean) instead of their full area is shown in Fig. 7. But aren't these percentages time dependent? This is problematic because the median curve is for the entire Cenozoic whereas the proportions are for the present-day. Not only has the area of the individual basins changed substantially since 66 Ma, but so has the pattern of sedimentation and lithology distributions. For example, the CCD has deepened substantially since 66 Ma, meaning that the global area available for carbonate sedimentation has also increased. As mentioned above, it would be helpful to have more information be supplied about what the deposition area through time actually looks like based on the authors' choices of paleo-topography maps and CCD changes. Have 'red clay' sections been removed from the LSR compilations in the 400+ drill sites? Following the logic above, failure to do so introduces a lithological bias, if not for the present-day then for some other time in the Cenozoic. We know that the CCD has deepened significantly since 55 Ma (e.g., Palike et al., 2012) meaning that the accumulation rate of carbonate was a lot lower in the past thanks to dissolution, and that sedimentation was dominated by much more slowly accumulating pelagic clay. So older sections could simply correspond to more slowly accumulating sediments rather than a drilling bias. This point needs to be clarified and some of these ideas explored in more depth. A map showing the distribution of drill sites should be included in the manuscript.

Indeed the size of the red clay free areas, as well as the basin areas, are time dependant. We thus recalculated this, while accounting for the changes in basin areas and changes in CCD per basins. This is what the new Figure 9d is.

> Line 104: The Pälike et al. CCD for the equatorial Pacific is not representative of the global ocean. This is an upwelling region of high productivity, which is strikingly different (much deeper in terms of CCD) from the rest of the Pacific, and most of the Atlantic and Indian oceans. The authors should consider using a more appropriate model or models (e.g., global curve of Boss and Wilkinson, 1991), that also cover the entire Cenozoic. Line 157: CCD trends in other ocean basins do not follow the trend for the Pacific CCD. This is evident in Campbell et al. (2018) and in Dutkiewicz and Muller (2022). The differences are quite striking. The statement should be corrected.

Indeed, this was an issue. We redid the full computation using basin-dependant estimates of CCD changes. This is what the new Figure 10 is showing.

Other comments Please refer to in-text figures in correct order. For example, Line 59 refers to Fig. 2 then Line 63 refers to Fig. 7. Fig.1 is first cited on Line 79 and there is no mention of Figs 3–6 prior to Line 63. This needs to be fixed as it's greatly reducing the readability of an already complex manuscript.

Please improve the readability of some of the figures. In many cases the fonts are too small (Figs 2 and 5, in particular).

All figures were redone.

Line 159: "Global mean deep sea pelagic sediment accumulation rates have decreased slightly during the Cenozoic." Relative to what? Please clarify.

Relative to their own past values.

Line 60: change "representative of pelagic sedimentation" to "representative of biogenic pelagic sedimentation".

Done.

Line 133 onwards: state which figure this text is referring to.

Done.

Line 175: "Our results thus largely reconcile the discrepancy that until now has existed between 87Sr/86Sr estimates of Cenozoic weathering, and the deep sea pelagic accumulation rate record." Please demonstrate this using your data and the seawater Sr curve. The curve will also provide an independent check of whether an increase in deep-sea accumulation rates corresponds to an increase in continental weathering.

A new figure was added to show this.

**Reviewer 2**

-The inter- vs intra-hole variability is a key finding, and the argument for a drilling bias hangs on it: if higher sedimentation rates nearer the modern do not relate to higher sedimentation rates earlier in the Cenozoic within each hole, then the drilling bias explanation doesn't hold. I think the overall thesis of the paper would thus be made stronger by exploring this finding more, and by clarifying the (currently very confusing) part on 'by-section normalization'.

-The use of red-clay-free area is an interesting new approach, but I have the same question as the first reviewer—what about the fact that the amount of red-clay-free has shifted over the Cenozoic? The maps in Wade et al. (2020), for example, show a much smaller red-clay-free area of seafloor in the early Cenozoic than later. Why is the basin median and mean calculated from the modern red-clay-free area, but then the CCD used in compiling the SAR curve? Although it would include a large unknown, estimating the red-clay-free area over time and using that estimate in the SAR calculations would likely be more accurate than assuming the same size of red- clay-area over time. Furthermore, if the same amount of carbonate and opal that is buried today were to be buried in the much smaller red clay free area of the early Cenozoic, apparent sedimentation rates would need to be substantially higher in the earlier Cenozoic than in the modern, no? Is that reflected in the raw LSR and SAR data? It isn't clear to me in Figure 1 that it is, but it's hard to tell from the way the data is depicted. Using the CCD combined with topographic data makes sense, and seems like a worthwhile approach, but I do wonder about opal sedimentation rates, particularly in the Southern Ocean. While these may be more minor in terms of global sedimentation rates, could they be useful in understanding at least the Southern Ocean history? A more detailed discussion of the topic would be helpful.

See comments for Reviewer 1.

-The manuscript emphasizes a very direct link between weathering and pelagic sedimentation, a claim that needs more support than it's given. Lines 39-41 say "If the deep sea pelagic sedimentary record is not (strongly) biased by hiatus-driven measurement scale artifacts, then the major increase in apparent rate over Cenozoic means either that geochemical proxies are somehow wrong and rate has changed exponentially, or that some other factors are biasing the Cenozoic deep sea record." The debate over the interpretation of those geochemical proxies is alluded to, but since the discrepancy between weathering and sedimentation rates is set up as the driving question this study is aiming to resolve, more

detail on the current state of that debate seems called for, including
more recent literature (e.g., Caves Rugenstein, Ibarra and von Blancken-
burg, 2019; Katchinoff et al. 2021; Pogge Von Strandmann, Kasemann
and Wimpenny, 2020). Line 175-176 makes the claim that "Our results
thus largely reconcile the discrepancy that until now has existed between
87Sr/86Sr estimates of Cenozoic weathering, and the deep sea pelagic ac-
cumulation rate record." This certainly needs a discussion of at least the
Sr record along with a visual comparison of the timeline of the strontium
data with the updated SAR curve. Is the EOT SAR step-change shown
in Fig. 8 reflected in the Sr data? What about the non-directional but
still substantial SAR variation in the Neogene?

We added large sections of text now discussing this in, hopefully, a clearer
and fairer way.

-Additionally, shallow marine settings are rejected as a possible alter-
native to pelagic settings for the burial of earlier Cenozoic weathering
products, but the two citations listed—particularly Ridgewell and Har-
greaves (2007)–do not by themselves provide an obvious explanation for
why this is the case. Elaboration would be helpful here, and updated
literature–e.g., for silica, Treguer et al. (2021), which substantially up-
dates the Treguer and de la Rocha (2013) estimates of shallow water Si
burial, and Rahman et al. (2017). (Although it would mean more work
and is perhaps beyond the scope of the paper, it would be interesting
to see a quantification of the magnitude of variation in shallow marine
carbonate and silica burial that would be required to accommodate a 2-
6x shift in pelagic sedimentation without a change in total weathering
product sequestration.)

We tried to address this in the introduction and the discussion. Though
to be fair it is not so much that we discard shallow marine settings than we
acknowledge that we simply do not have the data to assess them correctly and
that while they shouldn't be simply dismiss, we do have reasons to believe the
pelagic setting does account for much a significant enough proportion of the
weathering products to start drawing some conclusions based on it.

-Similarly, several authors have suggested that biogenic pelagic sedimen-
tation became more spatially concentrated over the Cenozoic (e.g., Dun-
lea et al 2017; Barron and Baldauf 1990). Some discussion of this hy-
pothesis and how this study's results fit in with it would be beneficial,
given that one of the exciting novelties of this work is its spatial speci-
ficity. Broadly speaking, to extend the conclusions of these analyses
beyond pelagic sedimentation (interesting in its own right) to summed

global weathering and make the claim that this solves a lynchpin of the Cenozoic weathering debate, there should be a more in-depth discussion of the abundant empirical (e.g., stable and cosmogenic isotopic, geophysical, terrestrial sedimentary, paleo-CO2) and model-based work that has been done on the subject and its conflicting interpretations, as well as alternative hypotheses.

We added large sections of text discussing empirical and model-based studies on weathering. Apart from the basin-based analysis, we did not go into the details of the spatial arrangements of biogenic pelagic sedimentation though as we felt it fell outside of the (already significantly expanded) scope of the MS.

Specific Comments As mentioned above, there are numerous minor errors and style issues throughout the text that, although I have not listed them here, should be cleaned up before publication. The abstract and the last paragraph of the introduction in particular need some serious textual work.

OK.

L42: the way 'decline' is used in the last sentence of the introduction, while correct, is confusing in the context of 'increase' in the paragraph before.

Reworded.

L55: Is it a running median with an overlapping 10 ky window, or a median (and mean) taken at 10 ky intervals starting centered at 5 ky? The former would make more sense for the purpose, I would think.

The data was binned in 10kyr intervals (thus non-overlapping).

L60: Given the wide range in sampling coverage between basins (12-220 sites), I wonder what a sort of rarefaction/bootstrap type analysis might show? In other words, what heterogeneity do you find if you subsample the Atlantic in groups of 12 sites with replacement? Just curious...

We reorganized the data in broader basins matching the CCD timeseries available: Pacific, Atlantic and Indian oceans (i. e. Southern Ocean is split between the three basins). The resulting datasets become more similar in size, hence subsampling would not be necessary but this was indeed a very good point. Unfortunately, because of time constraints, this is not an analysis we

were able to add to this resubmission.

> L63: Figure 7 is out of order, and Figure 1 isn't called out in the text until much later... Please order the figures correctly.

Corrected.

> L58: Figure 2 is cited here, but it doesn't do much to show how either LSR or SAR are calculated. It would be helpful to have a different figure showing the process and/or more clarifying text.

Corrected.

> Figure 3: Frequency would, I think, be more useful to the reader than density in these histograms. Also, panel labels (letters or numbers) here and throughout would make the figures easier to relate to the text and caption, rather than 'top left', 'bottom right', etc.

Panel labels were added.

> L77 cites a density plot of SAR vs time span in the Supplementary Material, but I can't see one there. Does it mean Figure 5?

Yes it is.

> L77: "We then calculate by-section normalized change in sedimentation rates vs geologic age in intervals with sedimentation to see if older time intervals, corrected for between site differences in local rates, show a decline in relative rate with increasing geologic age (see Figure 1)." I'm not sure I follow what was done here. Clarification of what is meant by 'by-section normalized' would be helpful.

We meant to say that each section's SAR/LSR was divided by its mean (i. e. the mean SAR/LSR of each of them becomes 1).

> L78-80: I'm confused by the wording here. What is being corrected for?

See above.

Figure 4: Hiatus frequency (bottom left panel) – does this refer to the proportion of the holes that include a given age and don't have sediment of that age? In other words, are they true gaps or do they include cores that simply don't reach the older ages? It would be helpful if this were made clearer in the caption.

Those are indeed true gaps: i. e. there are sediments recovered below the hiatus depth.

Figure 5: Very hard to read, should be cleaned up.

The figure was fully redone.

Figure 7 caption: It's unclear to me what 'area unweighted' means here. Is that corrected for CCD depth, red-clay-free area, basin area...?

The other curve being the area-weighted curve, the previous one (shown here for reference) was thus 'area-unweighted'. But fair enough this is a very awkward way of saying this and was thus modified.

L111-112: Figure 4 doesn't show that as far as I can tell – but it would be an interesting thing to show.

This is precisely what panel a is showing though.

L115: It seems from Figure 4 that hiatus frequency does decrease with age at both the start of the Cenozoic and approaching the modern, unless I'm missing something? It would also be useful to see a running median of hiatus length in the Figure 4 top left panel.

We now discuss this in the text (the modern decline is really just artifactual as hiatuses are mostly made by post-deposition erosional events).

Figure 1: It's almost impossible to see the black dotted line in the top panel, and there's no mention of it in the caption. This seems particularly important to show because the top and bottom panels actually look quite similar at first glance.

The line just represent "1", i. e. the mean of all the normalized LSR/SAR in each section. It is really just used as a grid line.

L130: Is that supposed to be a citation at the end of the sentence? Figure 6 is cited out of order. Please order the figures correctly.

Corrected.

**Reviewer 3**

The manuscript by Johan Renaudie and David Lazarus is an insightful study on how objective and pragmatic factors contribute to our understanding of how sediment accumulation rate estimates are biased with increasing geological time. It addresses the fundamental questions that paleoceanographic studies usually aim to answer, and thus it is very likely to become a highly-cited, influential paper. In my opinion, one important motivation for future readers of this article, will be pragmatic, i.e., what can I do to correct my data for the temporal bias in sedimentation rates? For this reason, I think it would be worthwile to include some kind of recommendations for future studies on how to handle the issues of compaction and drilling bias. Can a site or hole-specific correction factor be established from the data currently present in NSB, or included as supplementary data (see comment below)? Is it feasible/meaningful to compute such correction factor for individual sites? when global correction factor is applied, as in the figures in manuscript, but to a single site - will the results be meaningful? Speaking from my experience, I think expanding the discussion to include such issues would be highly beneficial for the readers.

This is now explicitly stated in the Discussions/Conclusions.

I do have several questions relating to the online supplementary materials. The supplement is supposed to include six items: the text, supplementary figures, Python code, and three additional data items. I failed to find the latter three. Were they included with the submission? I was also interested to actually run the code provided by the authors. The formatted text used in the supplement file, however, makes it difficult to copy the code to a code editor. As the authors apparently care for readers not familiar with programming (as revealed by the final part of the model description), this issue may be a nuisance to such readers. I also looked if the authors made the code available in an online repository (like the NSB code), but failed to find one.

Yes something went wrong with the SOM. A new one is being submitted and a repository is being set up.

---

## Referee Report (RR1)

Before I start, let me apologize for the tardiness of this review. Catching COVID for the first time at the beginning of the term hit me extremely hard, and I've spent most of my time trying to catch up.

I was very excited to be invited to review this paper. The authors have done multiple smaller studies to address fundamental questions about the marine sedimentological record and potential biases. They have examined the potential of hiatuses impacting sediment rates, age-related trends in rates, created a model to explore the above, then finally used the preceding results to inform a 'corrected' sediment accumulation rate.

I have only a few reservations which I believe could be handled in minor revisions. I do not believe there is need to perform additional analyses, but instead these are considerations which I think should be discussed and could be fodder for future studies. I am not sure how long this review is going to get, but that should be seen as interest and enthusiasm for publishing what is an excellent contribution, rather than critical flaws. I agree with the authors statement on lines 416-419, quantitatively establishing this as a fundamental bias in our marine sedimentological record, is of very broad significance.

My initial thought, back in the abstract actually, was "how are they going to address compaction?", and so I was suprised when there are only three mentions of compaction (one in fig. 7 caption, one when discussing erosion, and buried in the supplemental). I think this is the largest flaw here, we expect older intervals, usually buried under considerable mass from the sediments above, to have undergone compaction (water squeezed out, etc) - thus deeper sediments (typically older) should tend to have lower sedimentation rates when compared with those more shallow (thus younger). I would have expected this to be a discussion at the very least, or demonstrated that this isn't a feature of their "Within Section SAR vs. Time" analysis. Dealing with this does not need another analysis, but discussing how this impacts the results is certainly warranted.

Age model resolution - I was left with a question about the underlying resolution of the age models and how that changes through time. Being most familiar with the foraminifer biostratigraphic zonation scheme, the resolution is very different throughout the Cenozoic, with very short durations post KPg and in the latter portion of the Miocene and Plio-Pleistocene, and long zones in the Oligocene, for example. While that's certainly not 1 to 1 with the age models in NSB, I would imagine there are intervals which tend to have very highly resolved age models and those with less resolved. That might contribute to a few of these questions, though especially a trend in time with respect to the number of hiatuses. If there are intervals of time with poorly resolved age models, one would expect the probability of missing

a hiatus during that interval to be higher. I do think this is a finer point than it sounds, I generally agree with the statement made at 397-399 about age model quality, except that there could be pernicious systemic biases rather than the general uncertainty/diachroneity we usually worry about with biostratigraphy.

Sediment type - Probably my bias as a carbonate-focused worker, but I had also expected some discussion of two things: 1. paleoceanographers tend to fixate on finding considerable carbonate sediments due to their potential for lots of geochemical proxies, thus skewing records. 2. (and more importantly), there wasn't a lot of discussion about the different types of sediments found, other than distinguishing between clays below the CCD and carbonate above. Again, not suggesting to add an analysis of %carbonate or data from core description, but discussing the consequences (or lack there of) of different sediments in the potential environments seems valid.

Organizational - I was surprised by the "workflow" section, as it read to me as a methods+results summary prior to methods. I found it a bit jarring, but after coming back at the end of the paper, I understand it's likely there because this is essentially 4 small studies built together and either the authors or reviewers were expecting folks would get lost. I do, however, like the figure 2 associated with it quite a bit.

Caveats - Around L290 there's a discussion of the global pelagic biogenic sediment flux. I really like this section, but I would like there to be a short discussion of how to use those numbers or how to assess the uncertainty there. Given the analysis and uncertainties therein, should we only be interpeting the broad step around the EOT as real, should we be interpreting the broad Oligocene hump as real, or are the higher frequencies useful?

Supplemental files:
I have read the supplemental files, examined the figures. I am python illiterate (sadly, an R person only at the moment), so was not able to fully evaluate the code. I think I understand it somewhat, but without the skills to virtually-kick the tires that's the best I can do. I have not reviewed all the multitude of age models in SOM2, but NSB is the appropriate database to do this work on and these age models have been a part of many previous studies.

**Line - by line**

Line 40: I apologize for being a grammatical pendant but ending a sentence with "with" isn't appropriate, this should read: "There are, however, many general limitations with which studies of this type must deal."
41: first comma isn't neccessary.
49, 101: Earth should be capitalized.

124: SAR isn't defined yet (done on 137)

127,128: I bristled at the "typically with only a few, limited duration hiatuses." That's a statement they back up later, but I would have preferred to have a definition of what the authors were considering a hiatus (e.g., >0.5 myr).

145: Typo

147-149: I don't understand this sentence, and I'm not sure if it's because of missing words or what.

178: missing reference? "2001, ; geologically young"

179-180: You could probably show this quantitatively by plotting by expedition year, but I doubt that'd be interesting in this context (very interesting in others though!)

212-213: Isn't the 'unbiased selection' fairly subjective at this point?

248: space missing between sentences

255: typo

259: typo "insignicantly"

332-333: I think this sentence is too strong. I'd be ok if instead of "likely" the authors were to use "potentially", but I don't think the analysis here is specific enough about this question. It's certainly a testable hypothesis, however.

376: Typo "burg, 2010, , see Figure"

400: typo "documented. he one"

408-410: I don't know that this statement is true. I think that most of us are pretty aware that the record we have is biased in a tremendous number of ways, even just starting with IODP favoring carbonate sediments over others for paleoceanography, the loss of ocean crust through subduction leading to a tiny fraction of our exposed rocks being Cretaceous aged, and so on. Maybe it's because I'm an import from paleobiology (more-or-less), but I think this is accepted, but perhaps under discussed?

I, however, will defer to the authors with the phrasing here. This isn't a big deal.

---

## Author Response (AR2)

We hereby resubmit our manuscript 'Cenozoic pelagic accumulation rates and biased sampling of the deep-sea record' for your journal consideration. Apologies for the delay in sending this resubmission despite fairly minor revisions but in light of the broken agreement between Copernicus and the Leibniz-Insitut we had to make sure our institution will still cover the APC in case of acceptance, before doing so.

According to the last reviewer's comments, we added a note on compaction in-text and a figure showing the relationship between age model resolution and time in the SOM. Please find in the text below our answers to the reviewer's queries.

The authors,
Johan Renaudie and David Lazarus.

**Reviewer 1**

I was very excited to be invited to review this paper. The authors have done multiple smaller studies to address fundamental questions about the marine sedimentological record and potential biases. They have examined the potential of hiatuses impacting sediment rates, age- related trends in rates, created a model to explore the above, then finally used the preceding results to inform a 'corrected' sediment accumulation rate. I have only a few reservations which I believe could be handled in minor revisions. I do not believe there is need to perform additional analyses, but instead these are considerations which I think should be discussed and could be fodder for future studies. I am not sure how long this review is going to get, but that should be seen as interest and enthusiasm for publishing what is an excellent contribution, rather than critical flaws. I agree with the authors statement on lines 416-419, quantitatively establishing this as a fundamental bias in our marinesedimentological record, is of very broad significance.

Thanks a lot for your thoughtful review.

My initial thought, back in the abstract actually, was "how are they going to address compaction?", and so I was suprised when there are only three mentions of compaction (one in fig. 7 caption, one when discussing erosion, and buried in the supplemental). I think this is the largest flaw here, we expect older intervals, usually buried under considerable mass from the sediments above, to have undergone compaction (water squeezed out, etc) - thus deeper sediments (typically older) should tend to have lower sedimentation rates when compared with those more shallow (thus younger). I would have expected this to be a discussion at the very least, or demonstrated that this isn't a feature of their "Within Section SAR vs. Time" analysis. Dealing with this does not need another analysis, but discussing how this impacts the results is certainly warranted.

As explained to the previous reviewers, the concrete consequence of compaction is a change in the sediment density. As the computation for SAR is based on actually measured sediment density, and not an idealised version of it, compaction is in fact taken into account here. But indeed this explanation was missing in text, we have now added such a statement into the main text (lines 227–230).

Age model resolution - I was left with a question about the underlying resolution of the age models and how that changes through time. Being most familiar with the foraminifer biostratigraphic zonation scheme, the resolution is very different throughout the Cenozoic, with very short durations post KPg and in the latter portion of the Miocene and Plio-Pleistocene, and long zones in the Oligocene, for example. While that's

certainly not 1 to 1 with the age models in NSB, I would imagine there are intervals which tend to have very highly resolved age models and those with less resolved. That might contribute to a few of these questions, though especially a trend in time with respect to the number of hiatuses. If there are intervals of time with poorly resolved age models, one would expect the probability of missinga hiatus during that interval to be higher. I do think this is a finer point than it sounds, I generally agree with the statement made at 397-399 about age model quality, except that there could be pernicious systemic biases rather than the general uncertainty/diachroneity we usually worry about with biostratigraphy.

This is a fair point. We have thus added a figure in the SOM testing this issue. From what we know of NSB, while this issue would certainly be an issue in the Cretaceous, it is not, as the figure shows, an issue in the Cenozoic.

Sediment type - Probably my bias as a carbonate-focused worker, but I had also expected some discussion of two things: 1. paleoceanographers tend to fixate on finding considerable carbonate sediments due to their potential for lots of geochemical proxies, thus skewing records. 2. (and more importantly), there wasn't a lot of discussion about the different types of sediments found, other than distinguishing between clays below the CCD and carbonate above. Again, not suggesting to add an analysis of %carbonate or data from core description, but discussing the consequences (or lack there of) of different sediments in the potential environments seems valid.

Our paper is focussed on authigenic pelagic sediment. There are only three types that form substantial amounts of the deep sea sediment record – in pure form are: carbonate sediment, biosiliceous sediment and red clay. While red clay provinces are geographically large the average rates of sedimentation there are extremely low and these provinces thus contribute little to overall rates of sequestration. Much of the clay or fine silt in them also is from eolian transport of physical weathering products from land – the authigenic rates of sequestration are even lower than the bulk accumulation rates. Pure biosiliceous sediment is fairly uncommon in the deep sea, due presumably to the substantial amounts of weathering on land of carbonate rocks in addition to the more 'balanced' silicate minerals. Most biosilica is in sediments that also have significant, or are dominated by carbonate. Thus most of the biosilica record comes along automatically as the carbonate dominated sediment they occur in is selected by carbonate focussed drilling plannners. Some of the pure biosilica sediments have also been drilled. It is true that this last type of sediment has been to some degree undersampled by drilling (as radiolarian/diatom micropaleontologists ourselves we are acutely aware of this!), but the amount of undersampling of this fairly rare sediment type is not all that large, and due to its rarity, not significant for our study. Lastly we know of no reason why any bias would have

changed in relative intensity with geologic age. We thus feel that our calculations and the trends we have found are not significantly affected by biases in sediment type in our data.

> Organizational - I was surprised by the "workflow" section, as it read to me as a methods+results summary prior to methods. I found it a bit jarring, but after coming back at the end of the paper, I understand it's likely there because this is essentially 4 small studies built together and either the authors or reviewers were expecting folks would get lost. I do, however, like the figure 2 associated with it quite a bit.

This was indeed a request from the previous reviewers.

> Caveats - Around L290 there's a discussion of the global pelagic biogenic sediment flux. I really like this section, but I would like there to be a short discussion of how to use those numbers or how to assess the uncertainty there. Given the analysis and uncertainties therein, should we only be interpeting the broad step around the EOT as real, should we be interpreting the broad Oligocene hump as real, or are the higher frequencies useful?

This is a very good question but we do not have an easy answer to this. We think to adequately answer this we would need to examine how many actual sections are being used to calculate the global rate for any given time interval, how many of these are coming from a particular geographic region, or from other subsets of the data that might contain a bias; and create simulations to see how much variation could be produced by the granularity of variability in the dataset. This is all do-able but also fairly complex, and would address only one source of possible bias – there may well be others we haven't thought of that could be affecting short term calculated rates. What we did however is test the effect of two different paleotopographic models, and what the results showed is that the first-order trend is the same but smaller amplitude versions are different enough not to be fully trusted. We thus have decided to be as conservative as possible and here only interpret the first order trends. Hopefully our work will stimulate follow-up studies where the reality of the finer scale features of our results can be confirmed or not, and if found to be real, appropriately interpreted.

> Line 40: I apologize for being a grammatical pendant but ending a sentence with "with" isn't appropriate, this should read: "There are, however, many general limitations with which studies of this type must deal."

Modified accordingly.

> 41: first comma isn't neccessary.

Done.

49, 101: Earth should be capitalized.

Done.

124: SAR isn't defined yet (done on 137)

Done.

127,128: I bristled at the "typically with only a few, limited duration hiatuses." That's a statement they back up later, but I would have preferred to have a definition of what the authors were considering a hiatus (e.g., > 0.5 myr).

145: Typo

Corrected

178: missing reference? "2001, ; geologically young"

It was just a LATEX oddity, which is now corrected.

179-180: You could probably show this quantitatively by plotting by expedition year, but I doubt that'd be interesting in this context (very interesting in others though!)

Indeed.

248: space missing between sentences

Corrected.

255: typo

Corrected.

259: typo "insignicantly"

Corrected.

332-333: I think this sentence is too strong. I'd be ok if instead of "likely" the authors were to use "potentially", but I don't think the analysis here is specific enough about this question. It's certainly a testable hypothesis, however.

This is a fair point. We changed for "potentially".

376: Typo "burg, 2010, , see Figure"

Same as above.

400: typo "documented. he one"

Corrected.

408-410: I don't know that this statement is true. I think that most of us are pretty aware that the record we have is biased in a tremendous number of ways, even just starting with IODP favoring carbonate sediments over others for paleoceanography, the loss of ocean crust through subduction leading to a tiny fraction of our exposed rocks being Cretaceous aged, and so on. Maybe it's because I'm an import from paleobiology (more-or-less), but I think this is accepted, but perhaps under discussed? I, however, will defer to the authors with the phrasing here. This isn't a big deal.

Reworded to:

Although paleoceanographers have long known that older sediments are less commonly recovered than younger ones (Lisitzin, 1996) they have, to our knowledge, largely assumed that this phenomenon has not in itself introduced any substantial bias into the recovered representation of ocean sediments.